# Feed efficiency and maternal productivity of *Bos indicus* beef cows

**Danielly Fernanda Broleze[1], Luana Lelis Souza[1], Mariana Furtado Zorzetto[1], Rodrigo Pelicioni Savegnago[1], João Alberto Negrão[2], Sarah Figueiredo Martins Bonilha[1], Maria Eugênia Zerlotti Mercadante**[1]*

**1** Instituto de Zootecnia (IZ), Centro Avançado de Pesquisa de Bovinos de Corte, Sertãozinho, SP, Brazil, **2** Universidade de São Paulo (USP), Faculdade de Zootecnia e Engenharia de Alimentos, Pirassununga, SP, Brazil

* mezmercadante@gmail.com, mercadante@iz.sp.gov.br

## Abstract

This study evaluated 53 primiparous cows (36.8±1.23 months old and 484±40.9 kg of body weight) performance tested (GrowSafe® System) from 22±5 to 190±13 days of lactation in order to obtain daily dry matter intake (DMI). The animals received a high-forage diet (forage-to-concentrate ratio of 90:10). Milk production of the cows was evaluated three times by mechanical milking and the energy-corrected milk yield (ECMY) was calculated. Energy status (through the indicators glucose, cholesterol, triglycerides, and β-hydroxybutyrate), protein status (indicators albumin, urea, and creatinine), mineral status (indicators calcium, phosphorus, and magnesium), and hormonal status (indicators insulin and cortisol) were estimated four times throughout lactation. The residual feed intake (RFI) of cows was calculated considering DMI, average daily gain (ADG) and mid-test metabolic weight ($BW^{0.75}$) obtained in early lactation (from 22±5 to 102±7 days), and the animals were classified as negative (most efficient) or positive RFI (least efficient). The RFI model explained 53% of the variation in DMI. The mean DMI, ADG, ECMY, and calf weight as a percentage of cow weight were 12.47±2.70 kg DM/day, 0.632±0.323 kg/day, 10.47±3.23 kg/day, and 36.6 ±5.39%, respectively. Negative RFI cows consumed 11.5% less DM than positive RFI cows, with performance and metabolic profile being similar to those of positive RFI cows, except for a lower milk protein content and higher blood cholesterol concentration. In conclusion, negative (most efficient) and positive RFI (least efficient) Nellore cows, fed an *ad libitum* high-forage diet, produced similar amounts of milk, fat and lactose and had similar subcutaneous fat thickness, weight, calf weight as a percentage of cow weight, and blood metabolite concentrations (except for cholesterol). Therefore, there are economic benefits to utilizing RFI in a cow herd since cattle had decreased DMI with similar overall performance, making them more profitable due to lower input costs.

**Data Availability Statement:** All relevant data are within the paper.

**Funding:** This work was supported by Sao Paulo Research Foundation (FAPESP, grant #2015/

02066-4) for financial support and for providing grant to MFZ (FAPESP, grant #2016/24423-6), and Coordination for the Improvement of Higher Education Personnel (CAPES, Finance Code 001) for providing grants to DFB, LLS, and RPS. The funders had no role in study design, data collection and analysis, decision to publish, or preparation of the manuscript.

**Competing interests:** The authors have declared that no competing interests exist.

## Introduction

Improving the feed efficiency of beef cows while maintaining productivity levels should improve the profitability of cattle producers by reducing cow feeding costs and, hence, the feed costs per kg of calf weight gain during the pre-weaning period [1]. Gibb and McAllister [2] estimate that an increase of 5% in feed efficiency could have a four-fold greater economic impact than daily weight gain. The evaluation of phenotypic variation in feed efficiency within dams of different breeds and in different environments is essential to the understanding the impact of using more efficient animals on reproduction and productivity [3], since a reduction in fertility and in maternal traits may nullifies the advantages of the use of animals that consume less feed [4].

In the most classical measures of feed efficiency, there is no distinction between the energy used for separate functions. Conversely, residual feed intake (RFI) is represented as the residuals from regression of intake on the various energy sinks [4]. Therefore, RFI could take into account the energy expenditure for maintenance and production, and because it has a biochemical bases, it would potentially be applicable to animals irrespective of age and physiological status [5].

Few studies have so far evaluated the feed efficiency of lactating *Bos taurus* beef cows [6–8] and even less information is available for *Bos indicus* [9]. The results obtained by Black et al. [8], Walker et al. [3] and Souza et al. [9] show that most efficient (negative RFI) and least efficient (positive RFI) cows produce similar quantities of milk, but the former consume lower amounts of dry matter per day. The aim of this study was to evaluate two groups of lactating Nellore cows, positive or negative residual feed intake, and the effect of feed efficiency class on performance and maternal traits of cows from calving to weaning.

## Material and methods

### Animals

The study was conducted at Instituto de Zootecnia, Centro de Pesquisa em Bovinos de Corte, Sertãozinho, São Paulo, Brazil (21˚10′S and 48˚5′W). All animal procedures were approved by the Ethics Committee on Animal Use of Instituto de Zootecnia (Protocol 243–17), Nova Odessa, São Paulo, Brazil.

Fifty-three contemporary Nellore cow-calf pairs born in two consecutive breeding seasons and reared in an extensive pasture system consisting of *Brachiaria brizantha* were evaluated. Cows born in 2013 (n = 27) and 2014 (n = 26) were submitted to fixed-time artificial insemination (FTAI) at 2 years of age (January/February 2016 and December 2016/January 2017, respectively) using semen from a Nellore bull. Before calving (55±20 days), the cows were transferred to a pen. The animals remained in the pen until calving and were divided into calving groups (two groups/year) according to the calving date. After calving (36.8±1.23 months of age), the cows and calves were identified with radiofrequency identification (RFID) ear tags and transferred to a collective pen measuring 4,200 m$^2$ and equipped with 10 electronic Grow-Safe System$^{®}$ feeders (GrowSafe Systems Ltd., Airdrie, Alberta, Canada), where they received feed and water *ad libitum*. The cows and calves remained together in this facility from 22±5 days post-calving until weaning of the calves (190±13 days post-calving).

### Diets and feed sample analysis

The diet (Table 1) was formulated to meet the requirements for maintenance, growth and lactation of primiparous cows [10] to provide a weight gain of 0.75 kg/day. The vitamin A, D, and E requirements of cows were supplied by intramuscular application of vitamin supplement

(5mL) every 75 days. The diet was offered twice a day (8 a.m. and 4 p.m.) and the amount of feed was adjusted daily to maintain about 10% of leftovers. Weekly samples of the ingredients were obtained for determining the dry matter (DM) content of the diet. The weekly samples of dietary ingredients were pooled into six monthly samples, and their chemical composition was analyzed (Table 1). Total digestible nutrients of the diet (TDN, %DM) was estimated as TDN = tdCP + tdNFC + dNDF + 2.25 x tdEE—$FM_{TDN}$, where tdCP, tdNFC, and tdEE are the truly digestible fractions of crude protein, non-fibrous carbohydrates and ether extract, respectively (% DM); dNDF is the digestible neutral detergent fiber (% DM); and 2.25 is the Atwater's constant to equalize lipids and carbohydrates [11]. TDN were converted to digestible energy (DE) and metabolizable energy (ME) using the NRC (1996) [12] equations: DE (Mcal/kg) = $0.04409 \times TDN$ (%) and ME (Mcal/kg) = $1.01 \times DE$ (Mcal/kg) − 0.45 [10]. Net energy for lactation ($NE_L$) was estimated using NRC (2001) [10]: $NE_L$(Mcal/kg) = $0.0245 \times TDN$ (%) - 0.12; and Net energy for maintenance ($NE_M$), and for gain ($NE_G$) were estimated using NRC (1996) [12]: $NE_M$ (Mcal/kg) = $1.37ME - 0.138ME^2 + 0.0105 ME^3 - 1.12$; and $NE_G$ (Mcal/kg) = $1.42ME - 0.174 ME^2 + 0.0122 ME3 - 1.65$.

## Average daily gain, dry matter intake and residual feed intake

The animals were weighed every 23 days without previous fasting as recommended by Archer et al. [13], totaling 9 weight recordings for cows and 8 recordings for calves. The cows were submitted to FTAI during the experimental period and those that became pregnant before weaning had their weights corrected for the estimate of the conceptus weight [14]. The latter

**Table 1. Ingredients and chemical composition of the diet.**

| Item | Diet proportion | | |
|---|---|---|---|
| Corn silage (% DM) | 90.34 | | |
| Soybean meal (% DM) | 8.51 | | |
| Mineral salt[a] (% DM) | 0.83 | | |
| Urea[b] (% DM) | 0.32 | | |
| | Chemical composition | | |
| | Corn silage | Soybean meal | Diet |
| DM (%) | 36.2 | 85.3 | 41.4 |
| Ash (% DM) | 3.45 | 6.62 | 3.67 |
| Crude protein (% DM) | 6.98 | 47.0 | 11.1 |
| NDF (% DM) | 53.7 | 34.5 | 51.3 |
| ADF (% DM) | 21.4 | 14.0 | 20.5 |
| Lignin (% DM) | 6.03 | 1.96 | 5.59 |
| Ether extract (% DM) | 3.56 | 2.13 | 3.39 |
| TDN (% DM) | 64.7 | 72.0 | 64.5 |
| Metabolizable energy (Mcal/kg) | 2.43 | 2.75 | 2.42 |
| Net energy for lactation (Mcal/kg) | 1.47 | 1.64 | 1.46 |
| Net energy for maintenance (Mcal/kg) | 1.54 | 1.82 | 1.54 |
| Net energy for gain (Mcal/kg) | 0.94 | 1.19 | 0.95 |

DM: dry matter; NDF: neutral detergent fiber; ADF: acid detergent fiber; TDN: total digestible nutrients.

[a]Composition: 8 g/day phosphorus, 17 g/day calcium, 6.5 g/day sodium, 2.2 g/day sulfur, 0.8 g/day magnesium, 360 mg/day zinc, 100 mg/day copper, 70 mg/day manganese, 8 mg/day cobalt, 8 mg/day iodine, and 1.8 mg/day selenium.

[b]Reforce N (Petrobras): 450 g/kg of N.

was estimated for the day of weighing considering the days of gestation according to pregnancy diagnosis and approximate date of conception, as conceptus weight (kg) = (average calf birth weight x 0,01828) x (e $^{((0,02 \times t)—(0,0000143 \times t \times t))}$), where e is the Euler constant, and t are the days of gestation. The average daily gain (ADG) of cows was calculated for the early lactation period (from 22±5 to 102±7 days of lactation): i = $\alpha$ + $\beta$*DOTi + $\varepsilon$i, where $yi$ is the cow's weight in the $i$[th] observation, previously adjusted for the conceptus weight when appropriate; $\alpha$ is the intercept of the regression equation and represents the initial weight; $\beta$ is the linear regression coefficient and represents ADG; $DOTi$ is the day on test in the $i$[th] observation, and $\varepsilon i$ is the random error associated with each observation. The mid-test metabolic weight ($BW^{0.75}$) of cows was also calculated for early lactation (from 22±5 to 102±7 days): $BW^{0.75}$ = [$\alpha$ + (ADG × 0.5 × DOT)]$^{0.75}$, where $\alpha$ is the intercept of the regression equation and represents the initial weight, and $DOT$ are the days on test.

The intake of cows and calves was measured and recorded daily by the GrowSafe® System (GrowSafe Systems Ltd., Airdrie, Alberta, Canada). To prevent cows and calves from feeding simultaneously, which would compromise the recording of individual intake, two feeders were reserved only for calves by reducing the space between the vertical bars of the feeders so that the cows did not have access. In the other eight feeders intended for cow feeding, a wooden board was placed horizontally, which prevented the calves from reaching the feed because of their shorter stature compared to their dams. Dry matter intake (DMI) was calculated for the early lactation period as the mean of valid days of feed intake previously multiplied by weekly DM content of the diet.

First, the RFI of cows was estimated for the early lactation period (from 22±5 to 102±7 days post-calving) as the difference between observed and predicted DMI, and the cows were classified into two classes: most efficient (RFI<0) and least efficient (RFI>0). Next, DMI, ADG, $BW^{0.75}$, and RFI were calculated for the entire lactation period (from 22±5 to 190±13 days). The RFI of cows was estimated as the difference between observed and predicted DMI. The predicted DMI (DMIp) was obtained using the following multiple regression model: DMIp = $\beta_0$ + $\beta_1$ADG + $\beta_2$BW$^{0.75}$ + $\varepsilon$, where $\beta_0$ is the intercept of the equation; $\beta_1$ is the regression coefficient of DMI on ADG; $\beta_2$ is the regression coefficient of DMI on $BW^{0.75}$, and $\varepsilon$ is the RFI. Although a DMIp for lactating cows should include the energy sinks as milk yield and fat thickness, the model without these effects was chosen (the RFI Koch's model) precisely to verify the differences in milk yield and fat thickness of negative and positive RFI.

The following equation was fitted to obtain early lactation DMIp (from 22±5 to 102±7 days post-calving): DMIp = $\beta_0$ + (1.605) × ADG + (0.1467) × $BW^{0.75}$ + error ($R^2$ = 53.3%).

## Milk production, subcutaneous fat thickness and the calf weight as a percentage of cow weight

The milk yield (MY) of the cows was measured by mechanical milking at 63±5, 84±5 and 152±5 days of lactation as described by Souza et al. [9], using a method adapted from Walker et al. [3]. The calves were separated from the cows at 8 a.m., and each cow was mechanically milked after intravenous administration of 2 mL oxytocin for complete milk removal from the four quarters. The milk was discarded. The cows were returned to the paddock with *ad libitum* access to diet, water, and salt, and remained separated from their calves for 6 h. The cows were milked again to obtain the milk yield over 6 h. The milk yield was multiplied by four to obtain the 24-h milk yield (MY). A milk sample was collected during each milking for the analysis of milk composition (fat, protein, lactose, and total solids). The MY was corrected for energy according to Lamb et al. [15]: ECMY = (0.327 × kg MY) + (12.95 × kg fat) + (7.20 × kg protein), where ECMY is the energy-corrected milk yield.

The subcutaneous fat thickness (SFT) of cows was obtained at 21±5, 82±5, 143±8 e 184±12 days post-calving at five anatomical sites, assessing lumbar (SFT1, SFT2) and pelvic (SFT3 to SFT5) region: 12th–13th rib fat thickness (SFT1), longitudinal across the 11th–13th rib which captures three sites of fat thickness (SFT2) [16], transverse plane of the flank (SFT3), median transverse plane from the hook bone to the tip of the pin bone (SFT4) [17], and rump fat thickness (SFT5) [16]. The measurements were made with a 401347 Aquila ultrasound apparatus (Pie Medical Equipment B.V., Maastricht, The Netherlands) equipped with a linear 3.5-MHz probe (18 cm).

The calf weight as a percentage of cow weight was calculated considering cow and calf weights recorded on the same day from the beginning to the end of the lactation (8 calf and cow weight records) [13]: BWCA/BWC = (BWCA/BWC) × 100, where BWCA is the body weight of the calf and BWC is the body weight of the cow.

## Blood plasma metabolites

Blood samples were collected from all cows at 15±5, 41±5, 62±5 and 120±7 days of lactation (samplings 1, 2, 3 and 4) before the morning meal. The samples were collected into vacuum tubes by puncture of the jugular vein with sterile needles. The tubes contained heparin (separation of plasma), fluoride (glycolysis inhibitor for glucose analysis), and no coagulant (separation of serum). The samples were centrifuged at 3,500 rpm for 15 minutes for the separation of blood serum and plasma and stored in a freezer at -4 to -10˚C.

The indicators of energy status of the animals were measured using commercial enzymatic kits for the analysis of glucose, cholesterol and triglycerides (LaborLab, Votuporanga, SP, Brazil) and ß-hydroxybutyrate (Randox Laboratories, Crumlin, UK). The interassay coefficients were 8, 10, 7.5 and 5% for glucose, cholesterol, triglycerides and ß-hydroxybutyrate, respectively, and the intra-assay coefficients were 3, 7.5, 6 and 3%. The indicators of protein status was determined using enzymatic kits for albumin and urea (LaborLab, Votuporanga, SP, Brazil) and creatinine (BioClin, Belo Horizonte, MG, Brazil). The inter- and intra-assay coefficients were, respectively, 7, 5.5 and 6% and 2.5, 2 and 3.5% for albumin, urea and creatinine. The indicators of mineral status were measured using enzymatic kits for the analysis of calcium, phosphorus and magnesium (LaborLab, Votuporanga, SP, Brazil). The interassay coefficients were 8, 6 and 10% for calcium, phosphorus and magnesium, respectively, and the intra-assay coefficients were 5.5, 5 and 8%. The indicators of hormonal status of the animals were measured using an immunoenzymatic kit for insulin and cortisol (Monobind, Lake Forest, CA, USA). The inter- and intra-assay coefficients were, respectively, 8 and 10% and 3.5 and 6% for insulin and cortisol. Metabolites and minerals were analyzed by a kinetic enzymatic method in a Cirrus 80 MB spectrophotometer (FEMTOM, São Paulo, SP, Brazil), and the hormone analyses were performed using an enzyme immunoassay (ELISA) in a Labsystems Multiskan MS reader (Thermo Fisher Scientific, Waltham, MA, USA).

## Statistical analysis

The effect of RFI class (most efficient, RFI<0; least efficient, RFI>0) estimated for early lactation (from 22±5 to 102±7 days post-calving) on the traits studied was evaluated by fitting regression models. The PROC GLM procedure (SAS Institute, Inc., Cary, NC, USA) was used to fit the following linear model: $y_{ijklm} = \alpha + \beta_k + \beta_k^2 + CG_l + C\_RFI_m + \beta_k \times C\_RFI_m + \varepsilon_{ijklm}$, where $yi$ is the $i$th record of cow $j$ (j = 1,. . ., 53) on day $k$ of lactation (linear and quadratic effect, k = 11,. . . ., 210 days), of the $l$th contemporary group (l = 1,. . ., 4), in RFI class $m$ (m = 1, 2) for trait $y$; $\alpha$ is the intercept of the regression equation; $\beta_k$ and $\beta_k^2$ are linear and quadratic regression coefficients on day $k$ of lactation; $CG_l$ is the fixed effect of the $l$th calving

group; $C\_RFI_m$ is the fixed effect of RFI class $m$; $\beta_k$ x $C\_RFI_m$ is the effect of the interaction between the linear regression coefficient on day $k$ of lactation and of RFI class $m$, and $\varepsilon ijklm$ is the error associated with each observation. The estimated curves were used for interpolation of the value to all days of lactation over the interval from 11 to 210 days for performance traits, from 53 to 162 days for milk production and milk components, and from 4 to 140 days for blood metabolites.

Spearman correlations of RFI and of the traits used for the calculation of RFI (DMI, $BW^{0.75}$ and ADG) were estimated between early lactation (22±5 to 102±7 days post-calving) and the entire lactation period (22±5 to 190±13 days post-calving). This procedure was also performed for ECMY and blood plasma metabolites.

## Results

The cows had an initial weight of 484±41 kg, and the DMI and ADG during early lactation were 12.4±1.48 kg/day and 0.632±0.323 kg/day, respectively. The feed efficiency of lactating cows, evaluated based on RFI, was estimated considering the lactation period from 22±5 to 102±7 days. The multiple regression model explained 53.3% of the variation in DMI. Among the total variation in DMI of cows, 26.9% was explained by $BW^{0.75}$, 18.6% by ADG, and 7.8% by calving group. The mean RFI was 0±1.013 kg DM/day, ranging from -3.19 to 3.40 kg DM/day. Twenty-five cows with negative RFI (-0.792±0.705 kg/day) and 28 with positive RFI (0.707±0.662 kg/day) were identified. The descriptive statistics for performance traits, milk yield and blood metabolites evaluated from 22±5 to 102±7 days of lactation (early period of lactation) and from 103±7 to 190±13 days of lactation are shown in S1 Table and S2 Table, respectively.

Table 2 shows the Spearman correlations between RFI and between the traits used to calculate RFI obtained from 22±5 to 102±7 days of lactation (early lactation) and from 22±5 to 190±13 days (entire lactation period). The correlations were significant and high (P<0.01) for RFI, DMI and $BW^{0.75}$, and significant and medium (P<0.01) for ADG. Spearman correlations of milk yield and blood metabolites between early lactation and entire lactation period were also high, excepting for ECMY and triglycerides, which were medium, and for calcium and magnesium, which were low (S3 Table).

The results of analysis of variance of cow and calf performance traits evaluated during lactation are shown in Table 3.

The effect of RFI class on the DMI of cows was significant (P<0.0001) (Table 3). More efficient cows consumed less feed throughout lactation (Fig 1). The DMI of most efficient cows (negative RFI) was 11.6 kg DM/day and that of least efficient cows (positive RFI) was 13.1 kg DM/day, i.e., more efficient cows consumed -1.5 kg DM/day (or -11.5%) than positive RFI cows. The $R^2$ of the model was low as there was no significant variation in DMI over the days of lactation, only a declining trend (P = 0.065). This was expected since the cows were not in

**Table 2. Spearman correlation coefficients of performance traits of cows between 22±5 to 102±7 days of lactation and 22±5 to 190±13 days of lactation.**

| Trait | Correlation (P-value) |
|---|---|
| RFI | 0.89 (<0.0001) |
| DMI | 0.82 (<0.0001) |
| $BW^{0.75}$ | 0.95 (<0.0001) |
| ADG | 0.68 (<0.0001) |

RFI: residual feed intake; DMI: dry matter intake; $BW^{0.75}$: mid-test metabolic weight; ADG: average daily gain.

**Table 3. Effect of days of lactation, RFI class and interaction between days of lactation and RFI class on cow and calf performance traits evaluated throughout lactation.**

| Trait | P value | | | | Regression coefficient DOL*C_RFI | | | |
|---|---|---|---|---|---|---|---|---|
| | DOL | DOL$^2$ | C_RFI | DOL*C_RFI | Negative RFI | Positive RFI | SEM | R$^2$ |
| DMI cow, kg/day | 0.0650 | - | 0.0001 | 0.5863 | -0.002 | -0.002 | 0.001 | 0.09 |
| DMI calf, kg/day | 0.0001 | - | 0.4356 | 0.5186 | 0.022 | 0.021 | 3.12$^{E-04}$ | 0.58 |
| MY, kg/day | 0.0001 | 0.0061 | 0.3010 | 0.9177 | -0.143 | -0.142 | 0.044 | 0.19 |
| Milk fat, % | 0.0002 | - | 0.2611 | 0.7573 | 1.16$^{E-04}$ | 9.84$^{E-05}$ | 3.95$^{E-05}$ | 0.09 |
| Milk protein, % | 0.0001 | - | 0.0012 | 0.1570 | 4.79$^{E-05}$ | 6.79$^{E-05}$ | 9.90$^{E-06}$ | 0.37 |
| Milk lactose, % | 0.0001 | - | 0.4729 | 0.7613 | -2.49$^{E-05}$ | -2.75$^{E-05}$ | 6.14$^{E-06}$ | 0.20 |
| ECMY, kg/day | 0.0103 | 0.0265 | 0.1513 | 0.9083 | -0.170 | -0.169 | 0.070 | 0.08 |
| SFT1, mm | 0.0001 | - | 0.6942 | 0.7972 | 0.018 | 0.019 | 0.003 | 0.48 |
| SFT2, mm | 0.0001 | - | 0.9699 | 0.8265 | 0.028 | 0.029 | 0.004 | 0.40 |
| SFT3, mm | 0.0001 | - | 0.2378 | 0.8265 | 0.031 | 0.032 | 0.003 | 0.48 |
| SFT4, mm | 0.0001 | 0.0170 | 0.6339 | 0.9487 | 0.078 | 0.078 | 0.017 | 0.51 |
| SFT5, mm | 0.0001 | - | 0.8071 | 0.8425 | 0.047 | 0.049 | 0.005 | 0.45 |
| BWC, kg | 0.0001 | - | 0.4987 | 0.7506 | 0.679 | 0.702 | 0.051 | 0.45 |
| BWCA, kg | 0.0001 | - | 0.6146 | 0.4014 | 1.012 | 0.980 | 0.027 | 0.87 |
| BWCA/BWC, % | 0.0001 | - | 0.8061 | 0.3077 | 0.154 | 0.147 | 0.005 | 0.84 |

DOL: days of lactation (linear effect); DOL$^2$: days of lactation (quadratic effect); RFI: residual feed intake; C_RFI: RFI class; DOL*C_RFI: days of lactation within RFI class; SEM: standard error of the mean; R$^2$: coefficient of determination; DMI: dry matter intake; MY: milk yield; ECMY: energy-corrected milk yield; SFT1: 12th-13th rib fat thickness; SFT2: longitudinal across the 11th-13th rib which captures three sites of fat thickness; SFT3: transverse plane of the flank; SFT4: median transverse plane from the hook bone to the tip of the pin bone; SFT5: rump fat thickness; BWC: cow body weight; BWCA: calf body weight; BWCA/BWC: calf weight as a percentage of cow weight

the rapid growth phase. On the other hand, the DMI of calves born to positive and negative RFI cows was similar throughout lactation (Fig 1). The model explained a large part of the variation in the DMI of calves since the animals were in the rapid growth phase, with a consequent increase of DMI during the pre-weaning period (Table 3). The onset of DMI in calves occurred at 35 days of age, while the DMI prior to this day was very low or zero for most calves.

There was a quadratic effect of days of lactation on MY and ECMY (Table 3). Milk yield decreased across lactation, while %F and %P increased during the period studied (Fig 2). The MY, %F and %L were similar, while %P differed between RFI classes. Negative RFI cows produced milk with 4.0% protein, while positive RFI cows produced milk with 4.2% protein during the lactation period. Like MY, ECMY decreased gradually from the beginning to the end

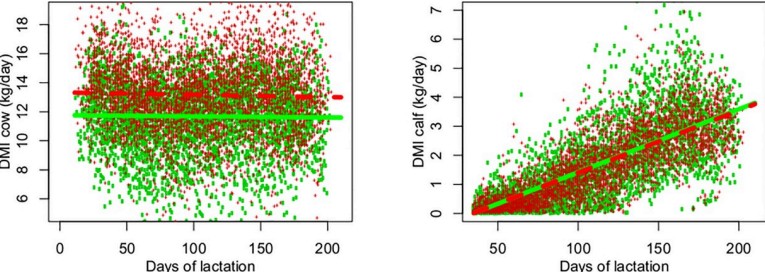

**Fig 1. Dry matter intake (DMI) of cows according to residual feed intake (RFI) class (left) and DMI of calves according to maternal RFI class (right) during lactation.** Observed (dots, ● negative RFI + positive RFI) and predicted (line, — negative RFI ---- positive RFI) values for negative (green) and positive RFI (red).

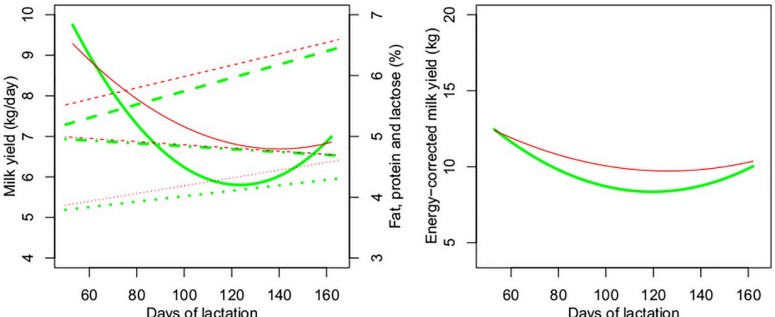

**Fig 2. Milk yield and milk fat, protein and lactose percentage (left, ─ milk yield ⋯⋯ protein ---- fat •‑•‑• lactose) and energy-corrected milk yield (right) of cows during lactation according to residual feed intake (RFI) class.** Predicted values (line, ─ negative RFI ─ positive RFI) for negative (green) and positive (red) RFI.

of lactation, without differences between RFI classes (Table 3 and Fig 2), despite the higher %P found in positive RFI cows.

The body condition of the cows, evaluated by SFT, was similar between RFI classes throughout lactation. There was a linear effect of days of lactation on the five anatomical sites evaluated, with a constant increase in fat thickness across lactation (Table 3 and Fig 3). Cows increased their weight considerably from the beginning to the end of lactation (Fig 3). A linear increase in calf weight was observed from the beginning to the end of the pre-weaning phase (Fig 4), as well as an increase in the calf weight as a percentage of cow weight (Fig 4). Cow (Fig 3) and calf (Fig 4) weights were similar between RFI classes throughout lactation (Table 3), as was calf weight as a percentage of cow weight (Fig 4).

Table 4 shows the results of analysis of variance of blood metabolites in cows during lactation. There was a linear effect of days of lactation on the blood concentrations of glucose, cholesterol, triglycerides, albumin, urea, creatinine, calcium, cortisol and insulin, as well as a quadratic effect on cholesterol and albumin. However, glucose, phosphorus and magnesium concentrations changed little during lactation (Figs 5 and 6). The concentrations of triglycerides (Fig 5), calcium and cortisol (Fig 6) decreased, and those of cholesterol, β-hydroxybutyrate, albumin, urea, creatinine (Fig 5) and insulin (Fig 6) increased during lactation. A significant difference in blood cholesterol concentration was observed between RFI classes (P = 0.012), with higher concentrations in negative RFI cows (204 mg/dL) compared to positive RFI cows (192 mg/dL) (Fig 5).

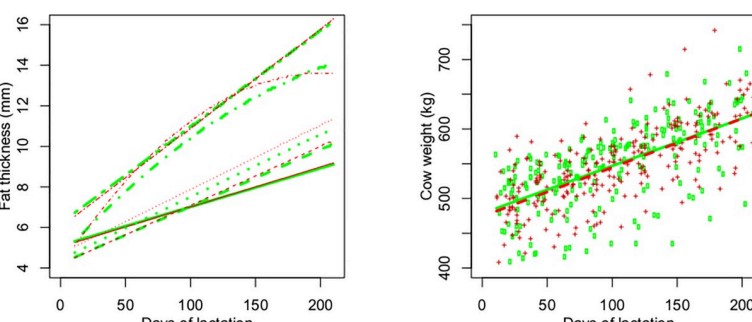

**Fig 3. Subcutaneous fat thickness of cows at five anatomical sites (left, ─ SFT1 ---- SFT2 ⋯⋯ SFT3 •‑•‑• STF4 •‑•‑• SFT5) and cow weight during lactation according to residual feed intake (RFI) class.** Observed (dots, negative RFI + positive RFI) and predicted (line, ─ negative RFI ---- positive RFI) values for negative (green) and positive RFI (red).

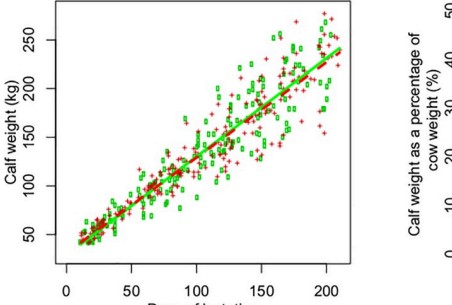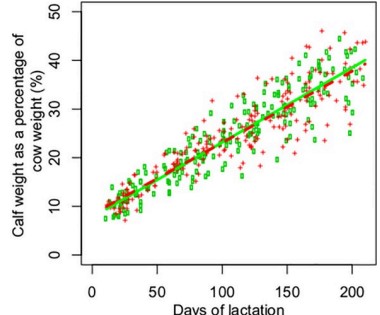

**Fig 4. Calf weight according to maternal residual feed intake (RFI) class (left) and calf weight as a percentage of cow weight (right) according to RFI class during lactation.** Observed (dots, ☐ negative RFI + positive RFI) and predicted (line, — negative RFI ---- positive RFI) values for negative (green) and positive RFI (red).

## Discussion

The present study compared cows that eat less (efficient) or more (inefficient), after accounting for ADG and BW$^{0.75}$, in terms of production (primarily MY, milk composition, calf weaning weight and calf weight as a percentage of cow weight) and metabolism (through the indicators of energy, protein, mineral and hormonal status). Studies suggested that the regression models used to predict RFI in growing cattle (the RFI Koch's model) may not be appropriate for lactating beef cows, as the majority of the phenotypic variance in DMI remains unexplained and/or the error in estimation of weight and weight gain is too high relative to that of DMI [18]. For lactating cows, RFI should represent the residuals of a multiple regression model of DMI on the main energy sinks (maintenance, body tissue mobilization, lactation, growth) [8, 4]. Although the RFI model for lactating cows should include the energy sinks, in the present study the model without these effects was chosen (Koch's model) precisely to verify the differences in milk yield and fat thickness of negative and positive RFI cows.

The regression model of DMI on ADG, BW$^{0.75}$ and calving group, adjusted for the calculation of RFI in Nellore cows of the present study, explained 53% of the variation in intake, a

**Table 4. Effect of days of lactation, RFI class and interaction between days of lactation and RFI class on blood metabolites evaluated during lactation.**

| Metabolite | P value | | | | Regression coefficient DOL*C_RFI | | | |
|---|---|---|---|---|---|---|---|---|
| | DOL | DOL$^2$ | C_RFI | DOL*C_RFI | Negative RFI | Positive RFI | SEM | R$^2$ |
| Glucose (mg/dL) | 0.0119 | - | 0.4112 | 0.5859 | -0.058 | -0.089 | 0.039 | 0.07 |
| Cholesterol (mg/dL) | 0.0001 | 0.0001 | 0.0120 | 0.1759 | 2.334 | 2.140 | 0.316 | 0.56 |
| Triglycerides (mg/dL) | 0.0001 | - | 0.6913 | 0.8302 | -0.095 | -0.086 | 0.029 | 0.16 |
| β-Hydroxybutyrate (mmol/L) | 0.7778 | - | 0.6135 | 0.7303 | 0.001 | 0.001 | 0.001 | 0.37 |
| Albumin (g/dL) | 0.0001 | 0.0062 | 0.1435 | 0.9228 | 0.028 | 0.029 | 0.006 | 0.55 |
| Urea (mg/dL) | 0.0001 | - | 0.5672 | 0.8065 | 0.450 | 0.470 | 0.059 | 0.44 |
| Creatinine (mg/dL) | 0.0001 | - | 0.1112 | 0.5901 | 0.004 | 0.005 | 0.001 | 0.49 |
| Calcium (mg/dL) | 0.1620 | - | 0.7071 | 0.5479 | -0.009 | -0.003 | 0.007 | 0.11 |
| Phosphorus (mg/dL) | 0.1692 | - | 0.6552 | 0.3389 | -0.001 | 0.003 | 0.003 | 0.42 |
| Magnesium (mg/dL) | 0.8721 | - | 0.9241 | 0.3428 | -0.001 | 0.001 | 0.002 | 0.004 |
| Cortisol (ng/dL) | 0.0001 | - | 0.2015 | 0.2093 | -0.104 | -0.177 | 0.041 | 0.14 |
| Insulin (ng/dL) | 0.1018 | - | 0.1404 | 0.9865 | 0.002 | 0.002 | 0.002 | 0.02 |

DOL: days of lactation (linear effect); DOL$^2$: days of lactation (quadratic effect); RFI: residual feed intake; C_RFI: RFI class; DOL*C_RFI: days of lactation within RFI class; SEM: standard error of the mean; R$^2$: coefficient of determination.

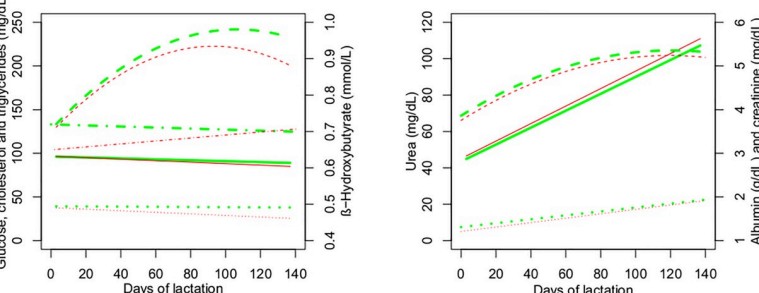

**Fig 5. Energy status (left, — glicose ---- cholesterol ········ triglycerides •-•-•- β-hydroxybutyrate) and protein status (right, — urea ---- albumin ······ creatinine) of cows during lactation according to residual feed intake (RFI) class.** Predicted values (line, — negative RFI — positive RFI) for negative (green) and positive (red) RFI.

percentage slightly lower than that reported by Black et al. [8]. These authors evaluated lactating *Bos taurus* beef cows receiving a forage-based diet and reported an $R^2$ of 60%, adjusting DMI for ADG, ECMY and rib fat thickness; however, surprisingly, $BW^{0.75}$ had no significant effect on the variation of cow DMI. In an extensive review, Kenny et al. [18] reported that $R^2$ is usually lower when the animals are fed forage-based diets because of the lower energy content of these diets and the lower rumen passage rate, reducing the expression of the DMI potential. In addition, the estimation of RFI is more complex in lactating cows compared to growing animals because the intake values and energy values for maintenance and production are highly variables. According to Kenny et al. [18], the average $R^2$ (70%) of RFI models in studies of young animals fed a high-concentrate diet is higher than the average $R^2$ (61%) in studies in which the animals receive a high-forage diet.

The RFI of the Nellore cows studied here ranged from -3.189 to 3.405 kg DM/day. Negative RFI cows consumed 1.5 kg DM/day less than positive RFI cows, corresponding to a reduction of 11.5%. Black et al. [8], studying taurine beef heifers after weaning and the same heifers during lactation, observed variations in RFI of -2.05 to 1.87 and -2.50 to 5.30 kg DM/day, respectively, i.e., the variation in RFI was much higher for lactating cows. The authors reported that low and medium RFI cows consumed 23.6% and 10.8% kg DM/day less than high RFI cows, and Walker et al. [3] reported a 6.5% lower DMI of cows with negative RFI compared to those with positive RFI.

The lactation curve of cows showed a declining trend and no peak lactation. The mean MY estimated was 7.59±2.17 kg/day and mean ECMY was 10.47±3.23 kg/day. Studies involving

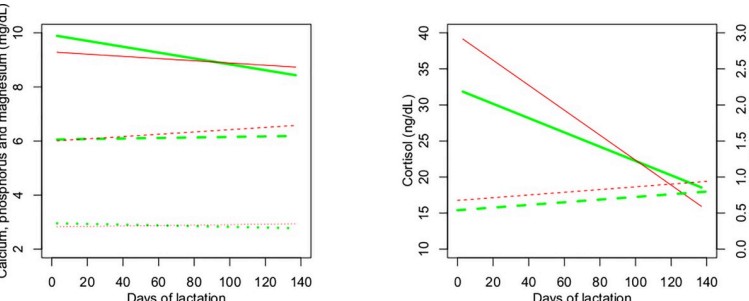

**Fig 6. Mineral status (left, — calcium ---- phosphorus ········ megnesium) and hormonal status (right, — cortisol ---- insulin) of cows during lactation according to residual feed intake (RFI) class.** Predicted values (line, — negative RFI — positive RFI) for negative (green) and positive (red) RFI.

Nellore beef cows reported lower uncorrected MY (3.16±0.31 and 3.70±0.33 kg/day) [19, 20] and corrected for 4% fat MY (7.0 kg/day and 7.2, 5.0 and 4.1 kg/day in early, mid and late lactation) [21, 22] than the means observed in the present study.

More and less efficient cows had similar MY and ECMY across lactation. Lawrence et al. [6] and Walker et al. [3] also found no relationship between uncorrected MY (obtained by the weigh-suckle-weigh technique) and RFI in *Bos taurus* cows, and cows classified as negative and positive RFI had similar MY [3]. Likewise, Black et al. [8] found no differences in ECMY of *Bos taurus* cows classified as low, medium and high RFI. Since the cows evaluated here are from a breeding program, they had maternal expected breeding value (EBV) estimated by maternal component of weaning weight, a proxy for milk yield in beef cows. Corroborating the previous results, the simple correlation between maternal EBV and the average of ECMY for entire lactation period was low but significant (0.2977, P = 0.0304), while simple correlations between maternal EBV and cow's RFI or RFI class for the early lactation period were 0.0819 (P = 0.5599) and 0.0888 (P = 0.5273). Therefore, there is no evidence that negative RFI cows produce less MY and ECMY than their inefficient counterpart.

The %F and %L were also similar in negative and positive RFI cows across lactation. However, the %P differed between RFI classes, with negative RFI cows producing milk with a lower %P than positive RFI cows. To improve energy balance is likely to have implications for more efficient animals, thus, a difference in some milk component between negative and positive RFI cows was expected since milk fat and protein synthesis represents a significant energetic expenditure for beef cows. In contrast, although Montanholi et al. [7] failed to establish any relationship between RFI and colostrum protein, fat, lactose or total solids percentages, the authors reported a negative correlation (−0.29) between RFI and milk lactose concentration. Although few and slightly contradictories, these results suggest that cow RFI may exert an effect on the milk composition.

The SFT obtained at five anatomical sites increased during lactation, and the cows did not mobilize reserves for maintaining milk production (negative energy balance). Fat thickness did not differ between cows with negative and positive RFI, in agreement with Lawrence et al. [6] and Black et al. [8] studying lactating beef cows. In growing animals, there are consistent results indicating a moderate genetic antagonism between SFT and feed efficiency, in which animals with a higher breeding value for fat thickness may be genetically less efficient [23, 24]. However, the relationship between fat deposition and feed efficiency in lactating cows is not well established.

Calves born to most and least efficient cows had a similar DMI from the feed. This finding indicates that, despite the lower milk protein percentage of negative RFI cows during lactation, the calves of these cows did not need to increase the intake of solid foods to compensate the lower amount of protein in the milk of their mothers. In addition, the pre-weaning weights of calves were similar for the most and least efficient cows. Calves born to cows with negative and positive RFI were weaned at an average weight of 226 and 221 kg, respectively (standard error of the mean = 6.52). No significant differences in the calf weight as a percentage of cow weight were observed between RFI classes, i.e., the most and least efficient cows produced a similar percentage of calf weight in relation to their own weight from the beginning to the end of lactation. At weaning, the calves weighed on average 36.6±5.39% of their mothers weight. Basarab et al. [14] also found no differences in the calf weight as a percentage of cow weight at calf weaning between cows classified as low, medium and high RFI. The authors reported a percentage (33.3%) similar to that observed in the present study.

Taken together, the results regarding the relationship between feed efficiency and calf weight as a percentage of cow weight agree with previous studies. Arthur et al. [25] described the relationship between feed efficiency and productivity of Angus cows after 1.5 generations

of selecting two divergent lines for low and high RFI (difference in the EBV of 0.80 kg DMI/ day). The pregnancy, calving and weaning rates, days to calving, calf weight per cow exposed, and milk production evaluated over three reproductive cycles were similar, with high RFI (most efficient) cows exhibiting more subcutaneous fat at the beginning of the breeding season. Morris et al. [26] also demonstrated that heifers born to sires with low EBV (most efficient) and high EBV (least efficient) for RFI did not differ in terms of pregnancy rate in the first or second breeding season, calf weight at birth or weaning, or milk production at 50, 100 and 150 days of lactation. These results encourage the use of negative RFI animals for sire and dam replacement in the herd, since these animals had decreased DMI with similar overall performance, making them more profitable due to lower input costs.

Nevertheless, RFI in lactating animals, although measuring feed efficiency per se, does not accurately reflect production efficiency. This is because the models used to calculate residual traits, as RFI, do not account for the partitioning of energy into the individual components, some of which are more economically important (e.g., milk fat and protein yield) than others (metabolic BW) [4].

Regarding blood metabolites, negative RFI cows had higher cholesterol concentrations than positive RFI cows during lactation. These results differ from those reported by Cônsolo et al. [27] who found lower plasma cholesterol levels in more efficient pregnant heifers, and by Wood et al. [28] who related low and nonsignificant correlation between plasma cholesterol levels and RFI or RFI class of mature pregnant beef cows. Although there is evidence of a positive relationship between RFI and plasma cholesterol in growing animals of some species as mice, pigs [29] and cattle [30], this relation in mature beef cows are not clear. In dairy cows, cholesterol metabolism is affected by energy deficiency depending on the stage of lactation. After 100 days of lactation, plasma cholesterol is increased in feed-restricted group of cows as a response to a negative energy balance [31]. Despite the fact that cows in the present study were fed for *ad libitum* intake, negative RFI cows consumed -11.5% DM than positive RFI cows. After the early lactation period plasma cholesterol increased in negative RFI cows (Fig 5), which was accompanied by a quadratic effect of DOL on SFT4 (subcutaneous fat thickness from the hook bone to the tip of the pin bone, Fig 3), albeit similar from both RFI classes.

There was no significant difference in blood glucose levels between the most and least efficient cows. The concentration of β-hydroxybutyrate was also similar between RFI classes, as well as the blood concentration of albumin, creatinine and urea. Although negative RFI cows consumed -11.5% DM than positive RFI cows, the amount of energetic substrate was sufficient to meet the nutritional requirements of the cows during lactation, which was evidenced by the ADG and increased subcutaneous fat thickness (Fig 3). Additionally, the plasma concentrations of β-hydroxybutyrate, creatinine and urea throughout lactation, together with the body weight and ADG similar for the two RFI classes, confirm the lack of mobilization of body tissues. This fact could be expected since the diet was formulated to support the requirements for growth, maintenance, pregnancy and lactation, allowing ADG of 0.750kg/day.

Blood calcium and phosphorus levels were similar in cows of the two RFI classes. Maintenance of blood calcium within the acceptable range of 8 to 10 mg/dl is a delicate balance between the demand for calcium for milk production and the homeostatic mechanisms of the cows to maintain blood calcium [32] (Fig 6). Cônsolo et al. [27] reported a trend (P = 0.06) toward higher calcium concentrations in more efficient pregnant heifers. These authors found higher phosphorus levels in more efficient animals and suggested greater availability of phosphorus for growth and energy metabolism.

Blood cortisol or insulin concentration did not differ between cows of the two RFI classes during lactation. One of the major biological responses to stress is the activation of the hypothalamic-pituitaryadrenal axis, which leads to the release of cortisol from the adrenal cortex

and the catabolism of energy stores to provide glucose. Cortisol production affects several metabolic and physiological processes, such as increased cardiovascular tone, and appetite modulation. Lactating animals display reduced neuroendocrine responses to hypothalamic-pituitaryadrenal axis activation compared with nonlactating animals [33]. Studying red and white blood cell parameters in steers genetically divergent for RFI, Richardson et al. (2002) [34] hypothesized that less efficient animals (high RFI) are more susceptible to stress than more efficient animals, and Richardson and Herd [35] observed in growing animals a trend towards lower cortisol concentration in negative RFI animals, indicating that more efficient animals are calmer and less reactive.

To mediate the nutrient fluxes towards the mammary gland for milk synthesis during early lactation, extensive endocrine changes coordinating homeorhesis are required. In particular, growth hormone concentration is elevated while insulin and IGF-I are low during the period of homeorhetic regulation of nutrient and energy partitioning to the mammary gland [31]. In lactating beef cows, Walker et al. (2015) [3] reported a positive correlation between insulin concentration and RFI. DiGiacomo et al. (2018) [33], in lactating dairy cows, observed lower cortisol response to adrenocorticotropic hormone and more responsiveness to lipolytic signals in low RFI cows compared to high RFI cows, suggesting that low RFI cows partition energy more readily away from storage in adipose tissue.

In conclusion, Nellore cows with negative (most efficient) and positive (least efficient) RFI that were fed a high-forage diet *ad libitum* produced similar amounts of milk, fat and lactose and had similar SFT, weight, calf weight as a percentage of cow weight and blood metabolite concentrations (except for cholesterol). Negative RFI (most efficient) cows had lower blood cholesterol concentrations and produced less milk protein, but their calves exhibited the same performance as those born to positive RFI cows, with the DMI of negative RFI cows being 11.5% lower throughout lactation. These results encourage the use of more efficient cows as replacement animals in the herd since they consume less feed without the loss of productivity, making them more profitable due to lower input costs.

## Supporting information

**S1 Table. Descriptive statistics for performance traits, milk yield and blood metabolites of Nellore cows evaluated from 22±5 to 102±7 days of lactation.**
(DOCX)

**S2 Table. Descriptive statistics for milk yield and blood metabolites of Nellore cows evaluated from 103±7 to 190±13 days of lactation.**
(DOCX)

**S3 Table. Spearman correlation coefficients of milk yield and blood plasma metabolites of cows between 22±5 to 102±7 days of lactation and 22±5 to 190±13 days of lactation.**
(DOCX)

**S4 Table. Pearson correlation among the components of feed efficiency with average of milk yield and blood metabolites evaluated from 22±5 to 102±7 days of lactation.**
(DOCX)

**S5 Table. Pearson correlation among the components of feed efficiency with milk yield and blood metabolites evaluated from 22±5 to 190±13 days of lactation.**
(DOCX)

## Author Contributions

**Conceptualization:** Maria Eugênia Zerlotti Mercadante.

**Data curation:** Luana Lelis Souza, Maria Eugênia Zerlotti Mercadante.

**Formal analysis:** Danielly Fernanda Broleze, Luana Lelis Souza, Rodrigo Pelicioni Savegnago, João Alberto Negrão.

**Funding acquisition:** Maria Eugênia Zerlotti Mercadante.

**Investigation:** Danielly Fernanda Broleze, Luana Lelis Souza, Mariana Furtado Zorzetto.

**Methodology:** Danielly Fernanda Broleze, Luana Lelis Souza, Rodrigo Pelicioni Savegnago, João Alberto Negrão.

**Project administration:** Maria Eugênia Zerlotti Mercadante.

**Resources:** Sarah Figueiredo Martins Bonilha.

**Supervision:** Mariana Furtado Zorzetto, Sarah Figueiredo Martins Bonilha.

**Visualization:** Danielly Fernanda Broleze, Luana Lelis Souza, Sarah Figueiredo Martins Bonilha.

**Writing – original draft:** Danielly Fernanda Broleze.

**Writing – review & editing:** Danielly Fernanda Broleze, Luana Lelis Souza, Maria Eugênia Zerlotti Mercadante.

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
