## [Decision Letter · Decision Letter 0]

5 Feb 2020

PONE-D-19-35367

Feed efficiency and maternal productivity of Bos indicus beef cows

PLOS ONE

Dear Dr. Mercadante,

Thank you for submitting your manuscript to PLOS ONE. After careful consideration, we feel that it has merit but does not fully meet PLOS ONE’s publication criteria as it currently stands. Therefore, we invite you to submit a revised version of the manuscript that addresses the points raised during the review process.

The manuscript is of interest for a great body of scientific field within the beef cattle industry. However, several concerns were raised by both reviewers and I concur with them. I strongly suggest to the authors to address to all comments made by the reviewers, which are critical points to better understanding of the results obtained as well as to improve the manuscripts' readability.

We would appreciate receiving your revised manuscript by March 25th. To enhance the reproducibility of your results, we recommend that if applicable you deposit your laboratory protocols in protocols.io, where a protocol can be assigned its own identifier (DOI) such that it can be cited independently in the future. For instructions see: http://journals.plos.org/plosone/s/submission-guidelines#loc-laboratory-protocols

We look forward to receiving your revised manuscript.

Kind regards,

Marcio de Souza Duarte

Academic Editor

PLOS ONE

Reviewers' comments:

Reviewer's Responses to Questions

**Comments to the Author**

1. Is the manuscript technically sound, and do the data support the conclusions?

Reviewer #1: Partly

Reviewer #2: Yes

2. Has the statistical analysis been performed appropriately and rigorously? 

Reviewer #1: Yes

Reviewer #2: Yes

3. Have the authors made all data underlying the findings in their manuscript fully available?

Reviewer #1: Yes

Reviewer #2: Yes

4. Is the manuscript presented in an intelligible fashion and written in standard English?

Reviewer #1: No

Reviewer #2: Yes

5. Review Comments to the Author

Reviewer #1: Efficiency in the cow herd is a very important topic to study. This study evaluates the relationship between RFI and millk characteristics and plasma metabolites hormones in bos indicus cows. As the authors pointed out, there has been limited research on this topic in bos indicus cows. The design of the experiment seems to be adequate. More detail is needed in the methods section, justification for using RFI as the efficiency measure is needed, and the discussion needs to be shortened, integrated better, and more emphasis placed on why the specific results were observed and what the results mean. See more below.

Line 28: What is meant by contemporary? include body weight

Line 36: These may be indicators of energy status but they do not measure energy status (same for protein, mineral, hormonal)

Line 46: More emphasis should be placed on the milk and plasma measurement results in the abstract. The other results are expected (eg. low rfi cows had lower DMI)

Line 50: What are the implications of the work?

Line 57: Change increasing to improving. Feed efficiency is a vague term and an increase could be a good or a bad thing so use the word improve or be specific in what feed efficiency measure you are talking about.

Line 60: this is what the reference reported of what they calculated. It doesn't mean that this will always be exactly a four-fold influence and it also depends on how you calculate it. Use caution in wording.

Line 65. Again, use caution in word. It could nullify any advantages but it depends on the magnitude of differences.

Line 74: There is a large amount of data on time of lactation effects. This is not mentioned in the objectives and it is not justified why this was done.

Line 84: Were they the same cows each year? This could influence how you approach the data analysis.

Line 87: Why does it matter if the bull is a low-rfi bull?

Line 88: What does collective mean?

Line 99: Why for a gain of 0.75 kg/d. I assume this is maternal ADG and not gain of cow with calf.

Line 106: Describe in more detail how TDN was estimated.

Line 123: These are approaches to estimate the weight of conceptus and associated fluids, tissues, etc. It should be indicated that it is an estimate and more detail is needed on how this was done. Also, is the ADG reported in the data this corrected ADG or actual. Was the corrected used for RFI calculations or actual? This needs to be more clearly presented.

Line 143: use RFI as the efficiency measure? RFI is not a good measure of efficiency in cows as in practice your goal is to feed for maintenance and not growth. If you have intake of cow and calf and weaning weight, why not examine effects on cow/calf efficiency (weaning weight divided by total intake of cow and calf) and considering BW change of the cow. Also, would it be useful to also examine relationships between the metabolites and milk data with the components of efficiency. Additionally, it would be more appropriate to analyze RFI as a continuous variable rather than breaking into 2 groups.

Line 160: More detail is needed on how the milking was done.

Line 172: Rather than say productive efficiency just spell out what it was weaning weight as a percentage of cow weight. What cow weight was used? The weight of the cow at calving or at a different time?

Line 184, 189, 192, 195: These are indicators and does not measure the status.

Line 204: Why were several approaches used. Could the same story be told with just 1 approach. The results and number of tables and figures could and should be reduced.

Line 227: why spearman correlations?

Line 245: Correlation of what? Should include p-values. Also, I am not sure that this information adds much as because of the approach used to calculate RFI you would expect there to be relationships between these variables. What is more unique is the plasma and milk data. The sections on other relationships could likely be deleted.

Line 254: Not sure this is the best way to present the data. Because much of it is just p-values it is difficult to understand the magnitude of the results.

Line 262: Not surprising. Could be deleted.

Line 279: Just discuss the higher order relationship - so just quadratic here.

Line 335: The discussion is quite long and discusses each result separately. It could be improved by better integrating the discussion to collectively come up with what you feel is most important. Also, the discussion largely focuses on comparing to other results rather than focusing on what is unique and what it means.

Line 339: bos taurus? be consistent with terminology

Line 342: Not sure that this is the reason why. Perhaps there are also differences in rumen capacity, etc.

Line 356: Maybe your data suggest this but it does not mean that it can be.

Line 359: Is this actual or maternal ADG. You formulated the diets for 0.75 kg ADG so maybe they gained less than expected.

Line 360: How did you formulate the diets. Did you formulate it for ad lib intake. If so, this is not a good discussion.

Line 366: This is a huge variation. Much of this discussion though is not needed

Line 379: If an objective was to examine milk characteristics throughout lactation this is ok to discuss but it should be clearly stated in the objectives.

Line 409: DMI from the feed or from feed and milk?

Line 434: Of course we would encourage using efficient animals. Maybe it suggests that selecting for RFI in the cowherd could improve production efficiency.

Line 441: Why are your results different. Is it something about bos taurus vs bos indicus? What does this mean?

Reviewer #2: General comments

The manuscript attempted to evaluate how feed efficiency in most and least efficient lactating Bos indicus cows, by means of residual feed intake, would correlate with the metrics associated with energy, protein, mineral and hormonal metabolism. The paper was cared for and thoroughly revised but some minor editing is still necessary. The introduction section needs to be focus on the topic at stake. Often times, the arguments are rather confusing to the reader.

Authors need to put a table with descriptive stats (mean, minimum, maximum) on variables of interest such as RFI, BW, ADG, DMI, MY, ECMY, SFT1, SF2, SFT3, SFT4, SFT5, glucose, cholesterol, triglycerides, beta-hydroxybutyrate, albumin, urea, creatinine, Ca, P, Mg, insulin and cortisol so the reader can have an idea on the magnitude of your data and whether or not values make sense.

The figures are not clear and need to be redesigned. Please take into account the color-blind reader as well as ones that print the paper with no color. Using distinct symbols and making the figures more clear will enhance the reading experience. Figures need substantial improvement.

Specific comments:

Line 28= Add “primiparous” after “contemporary”

Line 43 = Add “” after “efficiency”

Line 48 = Add comma “,” after cows

Replace “a high-forage diet ad libitum” by “and ad libitum high-forage diet”

Line 57 = why dairy has been brought up here? Suggest to introduce the topic related to your manuscript only so you don’t weaken the argument explored henceforth. This is a rather short space where authors should focus on the argument on lactation and how that is translated to efficiency of the pair cow-calf

Line 58 = Replace “cow feed ” by “feeding”

Line 59 = Replace “consequently, the feed costs per” by “hence”

Line 65-66 = Overall confusing introduction. Please explain what you mean by “maternal characteristics”. Did you mean mothering ability?

This whole section needs to be rewritten. I believe that point try to be made here is that consuming less feed is only advantageous if fertility and mothering ability are not jeopardized.

Line 71 = Add “than the latter” after “day”

Line 72 = Replace ”of lactating Nellore cows, and the effect of feed efficiency class” by “of two groups of lactating Nellore cows, positive or negative residual feed intake,”

Line 74 = Replace “birth” by “calving”

Delete “of their calves”

Line 82 = Add “born” after “pairs”

Replace “years of breeding” by “breeding seasons”

Line 87 = Why were the animals inseminated with only 1 bull? Could that potentially limit your inference given that only one father sired all the calves in this trial?

Lines 87-88 = the phrase “where they were fed silage” should be in another section “Diets and feed sample analysis”. Please remove from this section.

Line 90 = add “each group” after “age”

Line 99 = Why did you use NRC Dairy (reference #9) to formulate beef diets?

Line 100 = briefly describe the protocol of application for vitamin complex

Line 105 = Please call for Table 1 after “analyzed”

Line 109 = Why there is a call for BR-CORTE (reference #10) after the equation since reference was previously made (line 109) for NRC Dairy (reference #9)?

Line 111 = Table 1: Delete superscript #1 in the “Item” and at the Legend/footnote

Please describe the supplemented vitamin A/D/E supplement (concentrations for the intramuscular application)

Please consider adding information on NEm, NEg, NEl, and pregnancy to this table

Line 134 = See lines 92 and 93 for proper citation of GrowSafe

Line 140 = Replace “mothers” by “dams”

Line 160 = in brief describe your protocol to measure milk production (e.g.: frequency, how many quarts at the time, which quarts…)

Line 165 to 172 = please give the purpose for the 5 different SFT measurements and how these data were practically assessed by you.

Line 174-175 = Replace “calf body weight” by “is the body weight of the calf”

Replace “cow body weight” by “is the body weight of the cow”

Line 208 = I am not sure how SFT is different than SFT1 through SFT5. Please explain

Line 209 = Productive efficiency has already been defined on line 172. Please use acronym

Line 221 = italicize your beta so is standard throughout the manuscript

Line 237 = Where is this data?

Line 240 = Please delete “between” after “and”

Line 263 = How much less?

Line 264 = is 11.6 kg/d the average?

Line 270 = How much of the variation the model explained?

Line 272 = how is the onset of calf DMI intake being determined? What is the threshold? Please consider using supplemental DMI since the solids in the milk are also part of DMI and have been ingested since the beginning of lactation

Line 292-293 = How has body condition score been quantified and evaluated based on the ultrasound measurements of SFT (SFT1, SFT2,….SFT5)?

Line 344 = replace “complicated” by “complex”

Line 350 to 352 = I am not sure where are you going with your rationale here. Please complete your line of thought. Maybe explain why you should rely solely, or not, on RFI for your breeding program regarding your cows or if not too far, what about other efficiency indexes more suitable for inherent variation of this category.

Line 354 = Replace “days of lactation” by “DOL” since it has already been defined and is used in your tables.

Line 376 to 378 = Please discuss you argument based on what does that homogeneity of intake means…

Line 381 = Replace “of” by “involving”

Line 387 = Why? How would you justify no differences in milk yield (MY)? What were you EPD’s for MY?

Line 390 = Replace “milk fat” by “%F” since it has been defined on line 207

Replace “lactose percentages” by “%F” since it has been defined on line 208

Line 402= Replace “studied here” by “in this study”

Line 410 = Replace “milk fat” by “%F” since it has been defined on line 207

Line 422 = Why do you think that happened? Do you think the lack of differences associated with RFI limitations once evaluating productive efficiency

Line 416 = Replace “productive efficiency” by “PE” since it has been defined on line 172

Line 420 = Replace “productive efficiency” by “PE” since it has been defined on line 172

Line 427 = Add “(EBV)” after “value”

Line 435 = Complete your rationale…. Not more so on efficiency of the progeny but on decreasing feeding costs of the cows?

Line 437 = Please explain what is believed to be the reason for higher cholesterol found in more efficient cows?

Line 443 = What about yours? Why if you found quadratic effect on DOL fir cholesterol and SFT4? How lactation plays a role over the cholesterol metabolism?

Line 449 = Why wouldn’t you expect beta-hydorxybutyrate or glucose conc. differences for these growing animals? How would you explain this phenomena given the rate of growth of your animals and nature of your diet?

Line 458 = Please add information about your data to complete your rationale

Line 459 = What about Ca:P?

Line 464 = What about your cortisol? At any point, was cortisol ever an issue ? I didn’t see a big rationale toward this direction or a deeper application of it to your experimental goals.

Line 468-470 = What about yours? IF you found differences in DMI which led to differences in RFI, why there is no difference in insulin? Maybe a little discussion on how you believe your ad lib intake was controlled would enhance the quality of your findings.

6. PLOS authors have the option to publish the peer review history of their article (what does this mean?). If published, this will include your full peer review and any attached files.

Reviewer #1: No

Reviewer #2: No

---

## [Author Response · Author response to Decision Letter 0]

23 Mar 2020

Dear Dr. Marcio de Souza Duarte

Academic Editor – PLoS ONE

We would like to thank you, associate editor and reviewers, for your time and effort to improve the quality of the manuscript.

We tried to follow every recommendation of the two reviewers. 

Changes made to the previous version are in the “Revised Manuscript with Track Changes” file. 

Below are the individual answers to the reviewers.

Sincerely yours,

Maria Eugênia Mercadante, PhD

Instituto de Zootecnia

Reviewer #1

Re1. Efficiency in the cow herd is a very important topic to study. This study evaluates the relationship between RFI and millk characteristics and plasma metabolites hormones in bos indicus cows. As the authors pointed out, there has been limited research on this topic in bos indicus cows. The design of the experiment seems to be adequate. More detail is needed in the methods section, justification for using RFI as the efficiency measure is needed, and the discussion needs to be shortened, integrated better, and more emphasis placed on why the specific results were observed and what the results mean. See more below.

Au: More details were added in the methods section. A justification for using RFI as the efficiency measure was added in Introduction section (Macdonald et al., 2014). Finally, we change the Discussion section to be shorter and clearer.

Re1. Line 28. What is meant by contemporary? include body weight

Au: Contemporary means that the cows were born in the same herd and birth season, however the SD of the age already shows the contemporaneity of cows and it was deleted. Body weight was included.

Re1. Line 36. These may be indicators of energy status but they do not measure energy status (same for protein, mineral, hormonal)

Au: Ok. Some words were changed or/and included to change the meaning.

Re1. Line 46. More emphasis should be placed on the milk and plasma measurement results in the abstract. The other results are expected (eg. low rfi cows had lower DMI).

Au: The results of milk and metabolites have already been well emphasized in the abstract: “….with performance and metabolic profile being similar to those of positive RFI cows, except for a lower milk protein content and higher blood cholesterol concentration. In conclusion, ……., produced similar amounts of milk, fat and lactose and had similar SFT, weight, calf weight as a percentage of cow weight and blood metabolite concentrations (except for cholesterol).”

I agree the DMI differences between positive and negative RFI are expected, but I think it is still important to emphasize this results. 

Re1. Line 50. What are the implications of the work?

Au: Implications were include in the abstract.

Re1. Line 57. Change increasing to improving. Feed efficiency is a vague term and an increase could be a good or a bad thing so use the word improve or be specific in what feed efficiency measure you are talking about.

Au: Ok. It was changed.

Re1. Line 60. this is what the reference reported of what they calculated. It doesn't mean that this will always be exactly a fourfold influence and it also depends on how you calculate it. Use caution in wording.

Au: Ok, the sentence was changed to be less direct.

Re1. Line 65. Again, use caution in word. It could nullify any advantages but it depends on the magnitude of differences.

Au: Ok, the sentence was changed to be less direct.

Re1. Line 74: There is a large amount of data on time of lactation effects. This is not mentioned in the objectives and it is not justified why this was done.

Au: “From calving to weaning” gives the idea of time of lactation.

Re1. Line 84: Were they the same cows each year? This could influence how you approach the data analysis.

Au: No. Different cows were tested in each year (27 and 26 = 53 cows), all in their first breeding season at around 27 months of age, and all after their first calving.

Re1. Line 87: Why does it matter if the bull is a low-rfi bull?

Au: Ok. This information was removed.

Re1. Line 88: What does collective mean?

Au: Collective means that all cows were fed in the same feeder. However this was weird and was removed.

Re1. Line 99: Why for a gain of 0.75 kg/d. I assume this is maternal ADG and not gain of cow with calf.

Au: Ok, it was changed.

Re1. Line 106: Describe in more detail how TDN was estimated.

Au: More details of TDN estimation were included.

Re1. Line 123: These are approaches to estimate the weight of conceptus and associated fluids, tissues, etc. It should be indicated that it is an estimate and more detail is needed on how this was done. Also, is the ADG reported in the data this corrected ADG or actual. Was the corrected used for RFI calculations or actual? This needs to be more clearly presented.

Au: The paragraph was modified to include all modifications indicated above.

Re1. Line 143: use RFI as the efficiency measure? RFI is not a good measure of efficiency in cows as in practice your goal is on cow/calf efficiency (weaning weight divided by total intake of cow and calf) and considering BW change of the cow. 

Au: In relation to the use of RFI as a cow efficiency measure, I could agree with you. However, even though originally developed in growing beef steers, the international scientific literature has focuses on RFI as a efficiency measure for a variety of livestock, including cows (for example, dairy cows: Fischer et al., 2018-JDS, Seymour et al., 2020-JDS; dairy heifer: Green et al., 2013-JDS; Lage et al., 2019-PLoS One; ram lambs: Cammack et al., 2005-JAS; broiler chickens: Mebratie et al., GSE2019; pigs: Lu et al, 2017; beef heifers: Fitzsimons et al., 2013; besides results of beef cows cited in our manuscript: Lawrence et al., 211 and 2013, Walker et al., 2015; Montanholi et al., 2013; Black et al., 2013 ), but none of them reported results from Nellore cows. Due these facts, I do believe that our results will fill a lacuna in the feed efficiency of Bos indicus cows, and are worth publishing.

Re1. Also, would it be useful to also examine relationships between the metabolites and milk data with the components of efficiency.

Au: The relationship between variables was not explored as a result for two reasons: 1st) due to the repeated data of milk yield and its components (63±5, 84±5 and 152±5 days of lactation) and blood metabolites (15±5, 41±5, 62±5 and 120±7 days of lactation) throughout the lactation. Correlation analysis does not consider the covariance between repeated measures within animal, which results in somewhat weak results; 2nd) the design and the objective of the research was not to estimate the relations among these variables. However, I agree with you that these results can be of interest for the readers. So, 2 Tables with simple correlations (Pearson correlation) among the feed efficiency components (DMI, ADG, and BW0.75) from the early period versus MY and metabolites, and the same for the entire period of lactation were included in the Supporting Information (S4 and S5 Table).

Re1. Additionally, it would be more appropriate to analyze RFI as a continuous variable rather than breaking into 2 groups.

In relation of considering RFI as a continuous effect instead of 2 classes. We chose to analyze RFI as class because we study RFI throughout the lactation, and it could be difficult to interpret the interaction between 2 continuous effects (days of lactation X RFI). Besides this, RFI as class (low, medium and high; or negative and positive) is a well-established and accepted method, and the idea of RFI class (more and less efficient) has a more practical demonstration.

Re1. Line 160. More detail is needed on how the milking was done.

Au: Ok, more details about the milking method were included.

Re1. Line 172. Rather than say productive efficiency just spell out what it was weaning weight as a percentage of cow weight. What cow weight was used? The weight of the cow at calving or at a different time?

Au: Ok, the term “productive efficiency” was changed by “calf weight as a percentage of cow weight” throughout the text. “Calf weight as a percentage of cow weight” was calculated using 8 calf and cow records of weight, throughout the lactation. We included “….from the beginning to the end of the lactation (8 calf and cow weight records)…” to be clearest.

Re1. Line 184, 189, 192, 195. These are indicators and does not measure the status.

Au: Ok, we included “indicators of” in each line.

Re1. Line 204. Why were several approaches used. Could the same story be told with just 1 approach.

Au: We deleted “The following traits…….” to avoid repetition about the traits analyzed, as well as the explanation on order of the effects in the model. The other descriptions of the analyzes remained because they are important for complete understanding and for replication.

Re1. The results and number of tables and figures could and should be reduced.

Au: In the original version, carefully we decide to show p-values table and the corresponding figure that shows the average of the trait throughout the lactation by RFI class. Together, table and figure (some figure include observed data) show the variation of each trait, the trend during lactation, the difference between RFI classes and the statistical tests for days of lactation effect as well as for RFI class.

Sorry, but in the revised version we decided to keep all the tables and figures for the reason explained above. However, if the Editor-in-Chief decides that the figures and tables should be cut, we can do that in the next correction.

The figures were remade using the R software and we think they have become clearer.

Re1. Line 227. why spearman correlations?

Au: Spearman correlation measures the strength and direction of association between two ranked variables. We were interested in correlation between cows classification (ranking) based on early lactation and based on entire lactation period. Spearman correlation is recommended for correlation between two classifications.

Re1. Line 245. Correlation of what? Should include p-values. Also, I am not sure that this information adds much as because of the approach used to calculate RFI you would expect there to be relationships between these variables. What is more unique is the plasma and milk data. The sections on other relationships could likely be deleted.

Au: As pointed out above, we estimated the correlation between cows classification (ranking) based on a part of lactation period and cows classification (ranking) based on the whole lactation period. The p-values were included.

I am confident that these correlations (Table 2) are useful. Primarily because they are unique in Bos indicus cows, and even in Bos taurus cows (for instance, Black et al. 2013; and Walker et al. 2015, in Bos taurus beef cows, they measured DMI, ADG, RFI during part of lactation and do not even know if that part is representative of lactation as a whole). Furthermore, in dairy cows there are recent studies defining the optimal period length and stage lactation to estimate residual feed intake in dairy cows (for instance, Connor et al., J Dairy Sci. 2019) which shows the usefulness of these correlations.

In relation of milk and plasma correlation between the early lactation (22±5 to 102±7 days of lactation) versus the entire lactation period (22±5 to 190±13 days of lactation). A Table was included in the Supporting Information (S3 Table), the description of analysis was included in “Statistical analysis” and a commentary of the results was included in “Results”.

Re1. Line 254. Not sure this is the best way to present the data. Because much of it is just p-values it is difficult to understand the magnitude of the results.

Au: We chose to show all-important results, avoiding repetition. Carefully we decide to show p-values table and the corresponding figure that shows the average of the trait throughout the lactation by RFI class. Together, table and figure (some figure include observed data) show the variation of each trait, the trend during lactation, the difference between RFI classes and the statistical tests for days of lactation effect as well as for RFI class. It is the reason a table with descriptive stats of traits studied was not shown in the first version of the manuscript. In the corrected version, a Table with descriptive stats (mean, minimum, maximum) was included as Supporting Information (S1 Table and S2 Table).

The figures were remade using the R software and we think they have become clearer.

Re1. Line 262. Not surprising. Could be deleted.

Au: It was deleted.

Re1. Line 279. Just discuss the higher order relationship - so just quadratic here.

Au: Ok, it was corrected.

Re1. Line 335. The discussion is quite long and discusses each result separately. It could be improved by better integrating the discussion to collectively come up with what you feel is most important. Also, the discussion largely focuses on comparing to other results rather than focusing on what is unique and what it means.

Au: The Discussion section was changed.

Re1. Line 339. Bos taurus? be consistent with terminology

Au: Ok. “taurine” was change by “Bos Taurus”

Re1. Line 342. Not sure that this is the reason why. Perhaps there are also differences in rumen capacity, etc.

Au: It was not modified because it is a citation of Kenny et al. (2018).

Re1. Line 356. Maybe your data suggest this but it does not mean that it can be.

Au: Ok. “indicating” was changed by “suggesting”

Re1. Line 359. Is this actual or maternal ADG. You formulated the diets for 0.75 kg ADG so maybe they gained less than expected.

Au: This is maternal ADG (i.e., the cow body weight was corrected for the estimate of the conceptus weight before ADG calculation). For your recommendation this paragraph was removed.

Re1. Line 360. How did you formulate the diets. Did you formulate it for ad lib intake. If so, this is not a good discussion.

Au: Ok, I agree. The DMI and ADG of the cows were similar to that predicted in the formulation of the diet. This discussion is out of context and was removed.

Re1. Line 366. This is a huge variation. Much of this discussion though is not needed.

Au: Ok, we removed much of this discussion and left only two results for comparison.

Re1. Line 379. If an objective was to examine milk characteristics throughout lactation this is ok to discuss but it should be clearly stated in the objectives.

Au: Ok. “maternal traits” was included in the objectives to be more specific. “…of cows from calving to weaning” specify that the study included the entire lactation period.

Re1. Line 409. DMI from the feed or from feed and milk?

Au: We add “DMI from the feed”.

Re1. Line 434. Of course we would encourage using efficient animals. Maybe it suggests that selecting for RFI in the cowherd could improve production efficiency.

Au: Sorry, but I do not agree with the term “selection in cowherd” because selection is performed when the animals (young bulls and heifers) are chosen for replacement, before the reproduction. In the cowherd, the cows are discarded for low fertility or performance.

Re1. Line 441. Why are your results different. Is it something about bos taurus vs bos indicus? What does this mean?

Au: We could not find reasons for these differences. Kenny et al. (2018), in a comprehensive review, had emphasized the inconsistence in the literature for systemic metabolic indicators traits between negative and positive RFI cattle. 

Reviewer #2

Re2. The manuscript attempted to evaluate how feed efficiency in most and least efficient lactating Bos indicus cows, by means of residual feed intake, would correlate with the metrics associated with energy, protein, mineral and hormonal metabolism. The paper was cared for and thoroughly revised but some minor editing is still necessary.

Re2. The introduction section needs to be focus on the topic at stake. Often times, the arguments are rather confusing to the reader.

Au: Ok. The Introduction section was changed to be clearer.

Re2. Authors need to put a table with descriptive stats (mean, minimum, maximum) on variables of interest such as RFI, BW, ADG, DMI, MY, ECMY, SFT1, SF2, SFT3, SFT4, SFT5, glucose, cholesterol, triglycerides, beta-hydroxybutyrate, albumin, urea, creatinine, Ca, P, Mg, insulin and cortisol so the reader can have an idea on the magnitude of your data and whether or not values make sense.

Au: Ok. A Table with descriptive stats (mean, minimum, maximum) of all variables studied was included in the “Supporting Information” section (S1 Table and S2 Table).

Re2. The figures are not clear and need to be redesigned. Please take into account the color-blind reader as well as ones that print the paper with no color. Using distinct symbols and making the figures more clear will enhance the reading experience. Figures need substantial improvement.

Au: The figures were remade using the R software and we think they have become clearer.

Re2. Line 28. Add “primiparous” after “contemporary”

Au: Added.

Re2. Line 43. Add “” after “efficiency”

Au: The words were changed according the Reviewer 1.

Re2. Line 48. Add comma “,” after cows. Replace “a high-forage diet ad libitum” by “and ad libitum high-forage diet”

Au: Added and replaced.

Re2. Line 57. why dairy has been brought up here? Suggest to introduce the topic related to your manuscript only so you don’t weaken the argument explored henceforth. This is a rather short space where authors should focus on the argument on lactation and how that is translated to efficiency of the pair cow-calf

Au: “or dairy” was deleted.

Re2. Line 58. Replace “cow feed ” by “feeding”

Au: Replaced.

Re. Line 59: Replace “consequently, the feed costs per” by “hence”

Au: Replaced.

Re. Line 65-66. Overall confusing introduction. Please explain what you mean by “maternal characteristics”. Did you mean mothering ability? 

Au: “Characteristics” was changed for “traits”. I believe that “maternal traits” is more comprehensive than “mothering ability”. Even though “mothering ability” is a “maternal trait”, the first term does not seem to include milk quality, for example.

Re. Line 65-66. This whole section needs to be rewritten. I believe that point try to be made here is that consuming less feed is only advantageous if fertility and mothering ability are not jeopardized.

Au: Yes, it is what is written, according to Walker et al. (2015) and Berry & Crowley (2013).

Re2. Line 71 = Add “than the latter” after “day”

Au: “former” means “than the later”.

Re2.Line 72 = Replace ”of lactating Nellore cows, and the effect of feed efficiency class” by “of two groups of lactating Nellore cows, positive or negative residual feed intake,”

Au: Replaced.

Re2. Line 74 = Replace “birth” by “calving”. Delete “of their calves”

Au: Replaced and deleted.

Re2. Line 82. Add “born” after “pairs”. Replace “years of breeding” by “breeding seasons”

Au: Added and replaced.

Re2. Line 87. Why were the animals inseminated with only 1 bull? Could that potentially limit your inference given that only one father sired all the calves in this trial?

Au: The experiment was designed so that there was no interference of the sires breeding values for growth or RFI on the calf's performance (DMI and body weight), since the effect of RFI class can be small and difficult to detect in an experiment with relatively small number of animals. I do not believe this design limited the inference. We controlled one source of variation, which could be important.

Re2. Lines 87-88. The phrase “where they were fed silage” should be in another section “Diets and feed sample analysis”. Please remove from this section.

Au: Ok, it was removed.

Re2. Line 90. Add “each group” after “age”

Au: Sorry, I don’t think it is correct since it is the average of the two groups.

Re2. Line 99. Why did you use NRC Dairy (reference #9) to formulate beef diets?

Au: NRC Dairy was incorrectly cited. The diet was formulated using RLM 3.2 (https://www.integrasoftware.com.br/rlm31/produto.php), a version of the software for beef cattle diets. As the technology is based on equations developed by NRC (NRC. 1996. Nutrients Requirements of Beef Cattle. 7th rev. ed. Natl. Acad. Press, Washington, DC.), we referenced NRC 1996. The reference was changed.

Re2. Line 100. Briefly describe the protocol of application for vitamin complex.

Au: Ok, it was included.

Re2. Line 105. Please call for Table 1 after “analyzed”

Au: Ok, it was included.

Re2. Line 109. Why there is a call for BR-CORTE (reference #10) after the equation since reference was previously made (line 109) for NRC Dairy (reference #9)?

Au: NRC Dairy was incorrectly cited. The diet was formulated using RLM 3.2 (https://www.integrasoftware.com.br/rlm31/produto.php), a version of the software for beef cattle diets. As the technology is based on equations developed by NRC (NRC. 1996. Nutrients Requirements of Beef Cattle. 7th rev. ed. Natl. Acad. Press, Washington, DC.), we referenced NRC 1996. The reference was changed.

Re2. Line 111. Table 1. Delete superscript #1 in the “Item” and at the Legend/footnote

Au: Ok, it was deleted.

Re2. Line 111. Table 1. Please describe the supplemented vitamin A/D/E supplement (concentrations for the intramuscular application)

Au: It was included at line 100.

Re2. Line 111. Table 1.Please consider adding information on NEm, NEg, NEl, and pregnancy to this table

Au: Ok. NEl (NRC, 2001), NEm and NEg (NRC, 1996), were included included in Table 1, as well as in M&M.

Re2. Line 134. See lines 92 and 93 for proper citation of GrowSafe

Au: Ok, it was added.

Re2. Line 140. Replace “mothers” by “dams”

Au: Replaced

Re2. Line 160. In brief describe your protocol to measure milk production (e.g.: frequency, how many quarts at the time, which quarts…)

Au: The description was included.

Re2. Line 165 to 172. Please give the purpose for the 5 different SFT measurements and how these data were practically assessed by you.

Au: The purpose was to characterize the body condition at early, midlactation, and late lactation, using 2 sites proposed by Schwager-Suter et al., 2000 for dairy cows, and 3 sites proposed by BIF (2010) for beef cattle. They were obtained in 5 different body sites to assess the thickness of soft tissues in the lumbar (SFT1 and SFT2) and pelvic (SFT3, SFT4 and SFT5) region. Both publications bring schemes and images to recognize the body sites. After obtaining the images using Pie Medical ultrasound apparatus, they were “read” using the program Echo Image Viewer 1.0 (Pie Medical Equipment B.V., Maastricht, Netherlands, 1996). We modified the text to be clear. 

Re2. Line 174-175. Replace “calf body weight” by “is the body weight of the calf”. Replace “cow body weight” by “is the body weight of the cow”

Au: Ok, replaced.

Re2. Line 208. I am not sure how SFT is different than SFT1 through SFT5. Please explain.

Au: They were obtained in 5 different body sites to assess the thickness of soft tissues in the lumbar (SFT1 and SFT2) and pelvic (SFT3, SFT4 and SFT5) region. We can observe the differences in the Figure 3, and in Table S1 with descriptive stats (added as Supporting Information).

Re2. Line 209. Productive efficiency has already been defined on line 172. Please use acronym

Au: As suggested by Reviewer 1, “Productive efficiency” was change by “calf weight as a percentage of cow weight” itself, without acronym.

Re2. Line 221. Italicize your beta so is standard throughout the manuscript

Au: Ok, it was done.

Re2. Line 237. Where is this data?

Au: These percentages are from variance analyses to obtain predicted DMI from the model DMIp = β0 + β1ADG + β2BW0.75 + ε. We did not show a table for this variance analyses.

Re2. Line 240 = Please delete “between” after “and”

Au: It was not deleted because it could change the actual sense. We estimated the correlation between cows classification (ranking) based on a part of lactation period and cows classification (ranking) based on the whole lactation period for the same trait.

Re2. Line 263. How much less?

Au: The number are given right after, in the next phrase: “The DMI of most efficient cows (negative RFI) was 11.6 kg DM/day and that of least efficient cows (positive RFI) was 13.1 kg DM/day, i.e., more efficient cows consumed -1.5 kg DM/day (or -11.5%) than positive RFI cows.”

Re2. Line 264. Is 11.6 kg/d the average?

Au: Yes, this is the least square mean for “Negative RFI class”.

Re2. Line 270. How much of the variation the model explained?

Au: R2 were shown in Table 3 for performance traits and in Table 4 for blood metabolites. 

Re2. Line 272. How is the onset of calf DMI intake being determined? What is the threshold? Please consider using supplemental DMI since the solids in the milk are also part of DMI and have been ingested since the beginning of lactation.

Au: We could not consider the DMI before 35 days of age because the daily DMI was very low or zero, as explained in the text. 

As explained in M&M, the two performance tests started at 22±5 days on milk (or calves with 22±5 days of age). From 22 to 34 days, there were 573 records of DMI of the calves, which divided by 53 calves=10.8days/calf shows that some calves even were not in the facility (GrowSafe) every day. Analyzing these 573 records, 71% (n=406) had daily DMI<0.050 kg; 13% (n=77) had daily DMI from 0.050 to 0.100kg; 7% (n=38) had daily DMI from 0.100 to 0.200kg; 3.5% (n=20) had daily DMI from 0.200 to 0.300kg; 2,6% (n=15) had daily DMI from 0.300 to 0.400kg; and so on. Because this distribution, the model including DMI of the calves from 22±5 to 34±5 days of age did not fit well, giving non-sense results. 

Re2. Line 292-293 = How has body condition score been quantified and evaluated based on the ultrasound measurements of SFT (SFT1, SFT2,….SFT5)?

Au: Body Condition Scores (BCS) are based on visual and/or palpable assessment of the thickness of soft tissues in the lumbar and pelvic region. However, due to the subjective nature of BCS, their quality and repeatability has often been questioned. Nowadays, with ultrasound equipment BCS can be objectively assessed. Fat thickness of lumbar region was assessed by SFT1 and SFT2; and fat thickness of pelvic region was assessed by SFT3, SFT4 and SFT5.

Re2. Line 344. Replace “complicated” by “complex”

Au: Replaced.

Re2. Line 350 to 352. I am not sure where are you going with your rationale here. Please complete your line of thought. Maybe explain why you should rely solely, or not, on RFI for your breeding program regarding your cows or if not too far, what about other efficiency indexes more suitable for inherent variation of this category.

Au: Ok, some discussion about the use of RFI in cows was included at the beginning of Discussion section. Lines 350 to 352 were deleted.

Re2. Line 354. Replace “days of lactation” by “DOL” since it has already been defined and is used in your tables.

Au: Sorry but I thought the sentence would be unclear with this change.

Re2. Line 376 to 378. Please discuss you argument based on what does that homogeneity of intake means…

Au: Ok, a part of this paragraph was deleted. Two comparisons were maintained to show the consistency of our findings with the results of the literature.

Re2. Line 381. Replace “of” by “involving”

Au: Replaced.

Re2. Line 387. Why? How would you justify no differences in milk yield (MY)? What were you EPD’s for MY?

Au: We could not explain biologically the similar milk yield between negative and positive RFI cows, however this result was similar from all in the literature comparing milk yield and calf weight from negative and positive RFI cows (beef and dairy!!!). Since the cows evaluated here are from a breeding program, they have maternal EPD estimated by maternal component of weaning weight. The simple correlation between maternal EPD and ECMY22- 190 (energy-corrected milk yield from 22±5 to 190±13 days of lactation) was low but significant (0.2977, P=0.0304), while simple correlations between maternal EPD and RFI22-102 or RFI22-102 class were 0.0819 (P=0.5599) and 0.0888 (P=0.5273). These numbers show that our results are consistent. These results were included in the Discussion section, as well as more discussion.

Re2. Line 390. Replace “milk fat” by “%F” since it has been defined on line 207. Replace “lactose percentages” by “%L” since it has been defined on line 208

Au: Replaced.

Re2. Line 402. Replace “studied here” by “in this study”

Au: Replaced.

Re2. Line 410. Replace “milk protein” by “%P” since it has been defined on line 207

Au: Replaced.

Re2. Line 416. Replace “productive efficiency” by “PE” since it has been defined on line 172

Au: As suggested by Reviewer 1, “Productive efficiency” was change by “calf weight as a percentage of cow weight” itself, without acronym.

Re2. Line 422. Why do you think that happened? Do you think the lack of differences associated with RFI limitations once evaluating productive efficiency

Au: I think the “calf weight as a percentage of cow weight” index is one trait to compare productive efficiency of cows, among others as MY or milk composition. In my point of view, this index has limitations for comparing cows in a herd. Firstly, it is useful only to compare cows that calved an alive calf and weaned the calf; secondly, “productive efficiency” is highly influenced by cow age/parity (for instance, heifers will be always more productive than mature cows). In the present study we proposed to compare cows that eat more or less independently of their metabolic weight and ADG, in terms of production (mainly MY, milk composition, calf weaning weight and calf weight as a percentage of cow weight) and metabolism (through the indicators glucose, cholesterol, triglycerides, β-hydroxybutyrate, albumin, urea, creatinine, calcium, phosphorus, magnesium, insulin and cortisol).

Re2. Line 420. Replace “productive efficiency” by “PE” since it has been defined on line 172

Au: As suggested by Reviewer 1, “Productive efficiency” was change by “calf weight as a percentage of cow weight” itself, without acronym.

Re2. Line 427. Add “(EBV)” after “value”

Au: Added.

Re2. Line 435. Complete your rationale…. Not more so on efficiency of the progeny but on decreasing feeding costs of the cows?

Au: Ok, it was included

Re2. Line 437. Please explain what is believed to be the reason for higher cholesterol found in more efficient cows?

Au: Ok, the references were changed, and a possible explanation was included.

Re2. Line 443. What about yours? Why if you found quadratic effect on DOL for cholesterol and SFT4? How lactation plays a role over the cholesterol metabolism?

Au: Ok, a possible explanation was included.

Line 449. Why wouldn’t you expect beta-hydorxybutyrate or glucose conc. differences for these growing animals? How would you explain this phenomena given the rate of growth of your animals and nature of your diet?

Au: Ok, it was changed. An explanation was also included.

Re2. Line 458. Please add information about your data to complete your rationale

Au: It was deleted. These results were integrated above.

Re2. Line 459. What about Ca:P?

Au: Ca:P was also analyzed but it was not different from the analyses of Ca and P separately, so we did not include in the results.

Re2. Line 464. What about your cortisol? At any point, was cortisol ever an issue ? I didn’t see a big rationale toward this direction or a deeper application of it to your experimental goals.

Au: Ok, we include a discussion and we think it had improved.

Re2. Line 468-470. What about yours? IF you found differences in DMI which led to differences in RFI, why there is no difference in insulin? Maybe a little discussion on how you believe your ad lib intake was controlled would enhance the quality of your findings.

Au: ok, we added another reference.

---

## [Decision Letter · Decision Letter 1]

22 Apr 2020

PONE-D-19-35367R1

Feed efficiency and maternal productivity of Bos indicus beef cows

PLOS ONE

Dear Dr. Mercadante,

Thank you for submitting your manuscript to PLOS ONE. After careful consideration, we feel that it has merit but does not fully meet PLOS ONE’s publication criteria as it currently stands. Therefore, we invite you to submit a revised version of the manuscript that addresses the points raised during the review process.

We would appreciate receiving your revised manuscript by May 10th. To enhance the reproducibility of your results, we recommend that if applicable you deposit your laboratory protocols in protocols.io, where a protocol can be assigned its own identifier (DOI) such that it can be cited independently in the future. For instructions see: http://journals.plos.org/plosone/s/submission-guidelines#loc-laboratory-protocols

We look forward to receiving your revised manuscript.

Kind regards,

Marcio de Souza Duarte

Academic Editor

PLOS ONE

Reviewers' comments:

Reviewer's Responses to Questions

**Comments to the Author**

1. If the authors have adequately addressed your comments raised in a previous round of review and you feel that this manuscript is now acceptable for publication, you may indicate that here to bypass the “Comments to the Author” section, enter your conflict of interest statement in the “Confidential to Editor” section, and submit your "Accept" recommendation.

Reviewer #1: (No Response)

Reviewer #2: All comments have been addressed

2. Is the manuscript technically sound, and do the data support the conclusions?

Reviewer #1: Yes

Reviewer #2: Yes

3. Has the statistical analysis been performed appropriately and rigorously? 

Reviewer #1: Yes

Reviewer #2: Yes

4. Have the authors made all data underlying the findings in their manuscript fully available?

Reviewer #1: Yes

Reviewer #2: Yes

5. Is the manuscript presented in an intelligible fashion and written in standard English?

Reviewer #1: Yes

Reviewer #2: Yes

6. Review Comments to the Author

Reviewer #1: The manuscript has been improved. Thanks for the extensive revisions to improve the paper. I am OK with splitting the cattle into RFI groups but I do believe that it masks many of the effects and does not result int the data being used to its fullest extent. Yes it may be more practical to do this which may be more useful when presented to livestock producers but for scientific manuscripts analyzing data in a continuous fashion could result in a better explanation of the results (the shape of the curve of the relationship could be useful in understanding the physiology and for future prediction equations that could be developed). I will point out that just because there are many examples of publication breaking the data into RFI groups does not mean that it is the best way to analyze the data. I have a few other comments/suggestions for improvement. See below.

Line 75: This is an oversimplification of the concept of RFI. We have long known that there are many things besides BW that influence maintenance requirements. I would suggest rewording to be more careful with this interpretation of RFI.

Line 156: I think concept is supposed to be conceptus?

Line 388: I would suggest saying after accounting for ADG and BW rather than saying that it is independent. That is a very strong word and in actuality the traits are not independent - it is how the data is analyzed.

Line 470: change saved to improved energy balance (or something like that)

Line 519: This sentence is not clear. Please reword.

Line 530: Although there is evidence of a positive relationship between ...

Line 538: days of lactation

Line 540: Despite the fact that cows in the presebt styudt were fed for ad libitum intake, negative RFI cows ...

Line 557: This is highly dependent on the diet that the cows were fed. If I understand correctly, diets were formulated so that cows would be gaining maternal BW and BCS during the experiment. Because of this it may not be surprising that cows were not mobilizing body reserves. If a lower energy diet was fed, you likely would observe different results. This should be considered in this discussion.

Line 569: use alternative wording than balancing act

Line 583: need the last name of the author for the reference before 34

Reviewer #2: (No Response)

7. PLOS authors have the option to publish the peer review history of their article (what does this mean?). If published, this will include your full peer review and any attached files.

Reviewer #1: No

Reviewer #2: No

---

## [Author Response · Author response to Decision Letter 1]

28 Apr 2020

Dear Dr. Marcio de Souza Duarte

Academic Editor – PLoS ONE

We would like to thank you again, associate editor and #1 reviewer, for your time and effort to improve the quality of the manuscript.

We followed all recommendation of the Reviewer#1.

Changes made to the previous version are in the “Revised Manuscript with Track Changes-R2” file. 

Below are the answers to the Reviewer #1.

Sincerely yours,

Maria Eugênia Mercadante, PhD

Instituto de Zootecnia

Reviewer #1

Reviewer #1: The manuscript has been improved. Thanks for the extensive revisions to improve the paper. I am OK with splitting the cattle into RFI groups but I do believe that it masks many of the effects and does not result int the data being used to its fullest extent. Yes it may be more practical to do this which may be more useful when presented to livestock producers but for scientific manuscripts analyzing data in a continuous fashion could result in a better explanation of the results (the shape of the curve of the relationship could be useful in understanding the physiology and for future prediction equations that could be developed). I will point out that just because there are many examples of publication breaking the data into RFI groups does not mean that it is the best way to analyze the data. I have a few other comments/suggestions for improvement. See below.

Line 75: This is an oversimplification of the concept of RFI. We have long known that there are many things besides BW that influence maintenance requirements. I would suggest rewording to be more careful with this interpretation of RFI.

AU: We changed the sentences to give suggestive idea rather than affirmative one.

Line 156: I think concept is supposed to be conceptus?

AU: Yes. It was changed.

Line 388: I would suggest saying after accounting for ADG and BW rather than saying that it is independent. That is a very strong word and in actuality the traits are not independent - it is how the data is analyzed.

AU: Ok, it was changed

Line 470: change saved to improved energy balance (or something like that)

AU: Ok, it was changed.

Line 519: This sentence is not clear. Please reword.

AU: The first sentence was changed to be clear.

Line 530: Although there is evidence of a positive relationship between ...

AU: Ok, it was changed.

Line 538: days of lactation

AU: Ok, it was changed.

Line 540: Despite the fact that cows in the presebt styudt were fed for ad libitum intake, negative RFI cows ...

AU: Ok, it was changed.

Line 557: This is highly dependent on the diet that the cows were fed. If I understand correctly, diets were formulated so that cows would be gaining maternal BW and BCS during the experiment. Because of this it may not be surprising that cows were not mobilizing body reserves. If a lower energy diet was fed, you likely would observe different results. This should be considered in this discussion.

AU: We added a commentary about the diet.

Line 569: use alternative wording than balancing act

AU: Ok, it was changed by “delicate balance”.

Line 583: need the last name of the author for the reference before 34.

AU: It was already ok in the final R1 version.

---

## [Editor Report · Decision Letter 2]

15 May 2020

Feed efficiency and maternal productivity of Bos indicus beef cows

PONE-D-19-35367R2

Dear Dr. Mercadante,

We are pleased to inform you that your manuscript has been judged scientifically suitable for publication and will be formally accepted for publication once it complies with all outstanding technical requirements.

With kind regards,

Marcio de Souza Duarte

Academic Editor

PLOS ONE

Additional Editor Comments (optional):

The authors have adequately addressed all concerns raised in the last revision. The manuscript is ready to be further published. 
---

## [Editor Report · Acceptance letter]

20 May 2020

PONE-D-19-35367R2 

Feed efficiency and maternal productivity of *Bos indicus* beef cows 

Dear Dr. Mercadante:

I am pleased to inform you that your manuscript has been deemed suitable for publication in PLOS ONE. Congratulations! Your manuscript is now with our production department. 

With kind regards,

on behalf of

Dr. Marcio de Souza Duarte 

Academic Editor

PLOS ONE